# Surface Antifouling Modification on Polyethylene Filtration Membranes by Plasma Polymerization

**DOI:** 10.3390/ma13215020

**Published:** 2020-11-06

**Authors:** An-Li Hou, Szu-Yi Wang, Wen-Pin Lin, Wei-Hsuan Kuo, Tsung-Jen Wang, Meng-Jiy Wang

**Affiliations:** 1TaiDoc Corporation, B1-7F, No. 127, Wugong 2nd Rd., Wugu Dist., New Taipei City 24888, Taiwan; home00440@gmail.com; 2Department of Chemical Engineering, National Taiwan University of Science and Technology, 43, Keelung Rd., Sec. 4, Taipei 106, Taiwan; M9906064@mail.ntust.edu.tw (S.-Y.W.); D10306102@mail.ntust.edu.tw (W.-P.L.); D9506104@mail.ntust.edu.tw (W.-H.K.); 3Department of Ophthalmology, Taipei Medical University Hospital, Taipei Medical University, Taipei 110, Taiwan; 4Department of Ophthalmology, School of Medicine, College of Medicine, Taipei Medical University, Taipei 110, Taiwan

**Keywords:** non-fouling materials, ethylene oxide vinyl ether (EO1V), diethylene oxide vinyl ether (EO2V), polyethylene microfiltration membrane, plasma polymerization, surface modification

## Abstract

Surface modification on microporous polyethylene (PE) membranes was facilitated by plasma polymerizing with two hydrophilic precursors: ethylene oxide vinyl ether (EO1V) and diethylene oxide vinyl ether (EO2V) to effectively improve the fouling against mammalian cells (Chinese hamster ovary, CHO cells) and proteins (bovine serum albumin, BSA). The plasma polymerization procedure incorporated uniform and pin-hole free ethylene oxide-containing moieties on the filtration membrane in a dry single-step process. The successful deposition of the plasma polymers was verified by Fourier-transform infrared (FTIR), scanning electron microscopy (SEM), and X-ray photoelectron spectroscopy (XPS) analyses. Water contact angle measurements and permeation experiments using cell and protein solutions were conducted to evaluate the change in hydrophilicity and fouling resistance for filtrating biomolecules. The EO1V and EO2V plasma deposited PE membranes showed about 1.45 fold higher filtration performance than the pristine membrane. Moreover, the flux recovery reached 80% and 90% by using deionized (DI) water and sodium hydroxide (NaOH) solution, indicating the efficacy of the modification and the good reusability of the modified PE membranes.

## 1. Introduction

Filtration is a basic unit operation that separates permeate from retentate in mixed solutions through the filtration membranes [1,2]. For applications in biological processes, the recent advanced recombinant DNA technology involves recovery procedures for the production of proteins from mammalian cell culture systems by cell-protein separation through filtration. In particular, the separation of cells from mammalian cells through tangential flow filtration possesses advantages of high processing rates and maintaining cell viability [3,4]. The failure of the filtration membrane because of the reduction in flux usually results from the blockage of the pores of the filtration membrane due to the adsorption of solutes. Various methods have been reported to alter the surface of filtration membranes to become more hydrophilic for the improvement of fouling resistance. Polyethylene oxide (PEO) is the most commonly used molecule for surface modification in order to reduce the tendency of biomolecule adsorption due to the improvement in hydrophilicity [5,6], steric effect [7], and hydration layer [8].

The methods to incorporate ethylene glycol-containing moieties onto the surface of materials include simple immobilization [9], graft polymerization [10], and self-assembly of monolayers (SAMs) [11]. These methods suffer from either lacking stability or requiring multiple experimental steps or possible chemical residues. On the other hand, plasma treatment and plasma polymerization are dry processes that allow modifying surface of materials with fouling resistance. NH_3_ plasma was applied for the treatment of polypropylene (PP) membranes and the results show that 1 min treatment of PP improved the hydrophilicity and fouling resistance [12,13]. It has also been reported that plasma polymerized triglyme on polyamide membranes can improve fouling resistance, closely related to the pore size and surface roughness for the modified membranes [14,15].

In order to prepare non-fouling surfaces by overcoming the problems of lengthy experiments and solvent involvement, the present study employed plasma polymerization to deposit antifouling thin films using two types of EO-containing precursors—monoethylene vinyl ether (EO1V) and diethylene glycol vinyl ether (EO2V). The EO films were deposited by plasma polymerization under different applied power levels to evaluate the outcomes of biofouling toward proteins and mammalian cells. Moreover, the physico-chemical properties, including surface wettability, chemical functional groups, and surface charges of the deposited thin films, were evaluated to clarify the mechanism for fouling resistance.

## 2. Materials and Methods 

### 2.1. Materials and Chemicals

Polyethylene (PE) membranes with the average pore size of 7~12 μm were purchased from JL Special Material Group (Taipei, Taiwan). Ethylene oxide vinyl ether (EO1V, 97%, M.W.: 88.11 g·mole^−1^, boiling point (b.p.): 143 °C), diethylene oxide vinyl ether (EO2V, 98%, M.W.: 132.16 g·mole^−1^, b.p.: 196 °C), albumin from bovine serum (BSA, A7888-10G), and phosphate buffered saline (PBS, pH: 7.4, P3813-10PAK) were purchased from Sigma-Aldrich (St. Louis, MO, USA) and utilized as received without further purification. Pierce^®^ BCA protein assay kit (Bicinchoninic acid, BCA) was acquired from Thermal Scientific (Waltham, MA, USA). Sodium dodecyl sulfate (SDS) was obtained from Acros.

### 2.2. Plasma Polymerization

Plasma polymerization of EO1V and EO2V was conducted in a plasma system that was modified from the previous work [16]. In brief, the plasma system is composed of three main parts: (1) a reaction chamber, (2) a radio-frequency (RF) generator (Huttinger, model PFG 300 RF, Germany), and (3) a vacuum system. The schematic diagram of the plasma reactor is shown in Scheme 1. In the reactor, a stainless-steel anode with thickness of 2 cm was placed at the central part of the vacuum chamber, and its upper part was located 10 cm above the pumping outlet. The anode was separated by a 7 cm air gap from the shower-type cathode. Both electrodes were circular with diameter of 15 cm. The flow rate of the two precursors was calibrated prior to the experiments. The working pressure for the deposition of precursors was set at 100 mTorr, with flow rate of precursors at 20 standard cubic centimeters per minute (sccm), with a heating temperature of 100 °C, for 60 min. The generated plasma is a glow discharge. The vaporized precursor was introduced into the chamber and was polymerized with different applied power levels (10, 20, 30, 40, and 50 W) on the top of substrates including PE membranes (for filtration analyses), a KBr pellet (for infrared analyses), and an silicon (Si) wafer (for SEM). Due to overheating occurring at the applied power higher than 50 W for the plasma apparatus, the experimental parameter was controlled at the maximum applied power of 50 W.

### 2.3. Surface Characterizations

The morphology of pristine and plasma deposited PE was observed by using scanning electron microscope (SEM, JEOL JSM-6300, Akishima, Japan) and tapping-mode atomic force microscope (AFM, Digital Instruments, Nanoscope III, 125 μm AFM scanning head) with a scan resolution of 2 × 2 nm. The thickness of the deposited thin films was evaluated by α-stepper. The wettability of the surface was evaluated by measuring the static contact angles (Sindatek, Taipei, Taiwan) with deionized water. At least five droplets were measured for each position with a droplet volume of 3 μL. Surface charges of the samples were identified with an electrokinetic analyzer (BI-Eka, Anton Paar, Graz, Austria) at 25 °C and a pressure of 400 mbar. 

The chemical functionality of the plasma deposited membranes was characterized by Fourier-transform infrared spectrometry (FTIR), model FTS-3500 (Bio-Rad Digilab, Cambridge, MA, USA). The chemical composition of the plasma polymerized thin films was determined by electron spectroscopy for chemical analysis (ESCA). The Thermo VG Scientific Theta Probe Instrument (Waltham, MA, USA). with a monochromatic source of Al-Kα (1486.6 eV) as the excitation source was operated, with a pass energy of 50 eV. An Ar ion gun was employed with 3 kV voltage and 1 mA current. The characterizations of samples were taken under a takeoff angle of 53°, with an under X-ray spot size of 400 µm.

### 2.4. Zeta Potential Measurements

The zeta potential of the pristine and plasma polymerized PE was measured using the streaming potential method applied in an electrokinetic analyzer (EKA) (Anton Paar KG, Graz, Austria). The sample was equilibrated in a 1 mM KCl solution supporting electrolyte solution prior to ζ-potential measurements and the experiments were performed at constant temperature of 23–25 °C. The streaming potential is related to the zeta potential and was calculated based on the Fairbrother–Mastin equation. Equation (1) [17]:(1)$$\zeta =\frac{dU}{dP}\cdot \frac{\eta }{\varepsilon_{0}\varepsilon }\cdot \kappa \times 10^{-8}$$
where *ζ* is zeta potential (mV), *dU* is the difference in potential (mV/mbar), *η* is the viscosity of electrolyte (mPa.s), *ε*_0_ is the dielectric constant of vacuum (C^2^/Jm), ε is the dielectric constant of electrolyte (C^2^/Jm), and κ is the conductivity of electrolyte (mS/m).

### 2.5. Static Protein Adsorption

Bovine serum albumin (BSA) was chosen as a model protein for the evaluation of protein adsorption. First, 2 mg/mL BSA in PBS solution was prepared. PE membranes were incubated in protein solution for 24 h at 37 °C, and then washed with PBS and DI water five times to remove the weakly-bound proteins. Quantitatively, the bicinchoninic acid (BCA) protein assay kit was used to analyze the amount of adsorbed BSA where BSA was eluted from the sample by 1 wt % sodium dodecyl sulfate (SDS). The removed solution and BCA protein assay reagent were mixed in a ratio of 1:50. The absorbance was measured at wavelength 562 nm using a reader (iMark Absorbance Reader, BioRad, Hercules, CA, USA). A standard calibration curve was prepared based on the absorbance of a series of BSA concentrations.

### 2.6. Filtration Experiments

A crossflow filtration system was utilized to evaluate the resistance to fouling from Chinese hamster ovary (CHO) cells on pristine and modified PE membranes, acquired from Mycenax Biotech Inc., (Maioli, Taiwan). The system consists of an acrylic filter modulus with an active surface area of 25 cm^2^. The pristine and modified membranes were immersed in DI water for 15 min before filtration experiments, followed by filtrating cell solution under a fixed pressure of 8 psi and a flow rate of 3 ml/min. Cell solution was diluted until the optical density (O.D.) value equaled 0.3, prior to the filtration tests. For the filtration experiments, *J_p_* represents the flux after 1 min of filtration. The cake accumulated on the membranes was washed by flushing either with DI water or 0.1 N NaOH to compare the efficacy of cake removal. The second and third filtration tests were then performed and the flux for the first minute was represented as *J*_1_ and *J*_2_, respectively.

### 2.7. Statistical Analyses

Statistical analyses were performed by one-way analysis of variance (ANOVA) executed by Origin^®^ software. Tukey comparison tests were utilized to compare the differences among samples. In all tests, the significant level was indicated by the asterisk mark (*): * for *p*-value < 0.05, ** for *p*-value < 0.01, and *** for *p*-value < 0.001.

## 3. Results

The effects of plasma polymerization deposition on polyethylene membranes using two ethylene oxide-containing precursors, EO1V (ethylene oxide vinyl ether) and EO2V (diethylene oxide vinyl ether), were characterized by the deposited thickness, water contact angles (WCAs), zeta potential, and SEM. The filtration performance was evaluated by directly filtrating CHO cells. The surface fouling properties were quantified by the amount of adsorbed protein using bovine serum albumin (BSA). 

### Characterizations of pp-EO1V and pp-EO2V

The effects of plasma applied power on the deposited thickness of plasma polymerized EO1V and EO2V (pp-EO1V and pp-EO2V) were firstly evaluated. The thickness of the plasma polymer thin films on Si wafer was measured by an alpha-stepper and the results show that the thickness of pp-EO1V and pp-EO2V increased linearly to 9.2 nm and 13.6 nm for the applied power ranging from 0 to 50 W, respectively (Figure 1). The larger thickness of the deposited pp-EO2V when compared with that of pp-EO1V is presumably due to the longer chain length of the precursor of EO2V, which provides more plasma excited fragments for deposition (Figure 1(ii)) [18].

Moreover, the results of the WCA show that the deposition of EO1V and EO2V can bring hydrophilic properties to the substrate. Pristine Si wafer possesses a WCA of 58.6°. The deposition of both EO1V and EO2V altered the WCA toward hydrophilicity, so that the WCA was 18.6° on both pp-EO1V/PE and pp-EO2V/PE (Table 1). It is noted that the higher applied power showed no influence on the water contact angle, indicating that the surface wettability depends only on the outer most layer of the plasma deposited ethylene oxide-containing thin films that revealed hydrophilic property.

The variation of surface morphology was evaluated by plasma deposition of EO1V and EO2V on PE filtration membranes. The porous PE membrane clearly revealed the sintered particulate morphology (Figure 2a). The deposition of pp-EO1V and pp-EO2V on PE resulted in granular coatings when the applied power ranged from 10–30 W and became a layer of coating when the applied power was 40 W and 50 W (Figure 2b–k). In addition, AFM analyses revealed that the deposited EO1V and EO2V plasma polymers on Si were uniformly distributed with the values of surface roughness ranging from 0.23–0.36 nm (Appendix A, in Appendix A, and Table 1). No significant difference was found for the obtained roughness.

FTIR spectra of EO1V and EO2V plasma polymers deposited on PE filtration membranes at different applied plasma power levels revealed that hydroxyl groups (-OH) at 3400 cm^−1^ and between 1050–1300 cm^−1^, and carbonyl groups (C-O) at 1621 cm^−1^ were found (Figure 3). 

ESCA spectra analyses showed that pristine PE is composed of 97.2% carbon and 2.8% of oxygen. By depositing pp-EO1V and pp-EO2V on PE, more than 16% of oxygen was incorporated on the surface of PE (Table 2, and Appendix A). The increase in the plasma applied power resulted in increased oxygen content, so that the highest oxygen content reached 26.2% and 27.8% on pp-EO1V/PE and pp-EO2V/PE, respectively. The deconvolution of the high-resolution spectra revealed that the resultant oxygen-containing functionalities are mainly C-O and C=O, which also increased with the function of the plasma applied power. Both FTIR and ESCA results verified the successful incorporation of EO1V and EO2V via plasma polymerizations. 

The outcomes of plasma polymerization using EO1V and EO2V were evaluated by measuring the accumulated weight of the cell solution that passed through the filtration membranes, where larger flux indicates the membrane possessed a larger filtration capacity. On the pristine PE filtration membrane, the accumulated weight was 259.8 ± 16.2 g after 3 h filtration. On pp-EO1V/PE, the accumulated weight increased to 359.5 ± 13.9 g, 366.9 ± 18.9 g, and 371.6 ± 13.6 g on 10W-EO1V/PE, 20W-EO1V/PE, and 30W-EO1V/PE, respectively (Figure 4a). However, when the plasma applied power was further increased, the accumulated weight of the filtrates decreased slightly, so that 364.5 ± 12.1 g and 357.3 ± 14.9 g were found on 40W-EO1V/PE, and 50W-EO1V/PE. The results indicate that the plasma polymerization of EO1V at 30 W provided the best fouling resistance toward CHO cells and the filtration performance was improved by more than 1.43 fold compared to the pristine PE filtration membrane. Similarly, the highest accumulated weight on EO2V deposited PE was also found on 30W-EO2V/PE, which provided 1.45 fold higher filtrate than that on the pristine PE (Figure 4b). 

Meanwhile, the higher filtration flux (~221.84 L/m^2^·h) was found on pp-EO1V/PE and pp-EO2V/PE in comparison with that on pristine PE (174.40 L/m^2^·h). The filtration results clearly show the slower reduction and later blockage of EO1V/PE and pp-EO2V/PE when compared with the pristine PE (Figure 4c,d). It is noted that all the filtration results reached the near zero flux quantity after 40 min of filtration, indicating that the termination of the filtration and equilibrium filtered cakes were obtained, which is closely related to the amount of the cell solutions applied in this study. Due to the limit of the crossed-flow filtration system (as shown in Appendix A), the surface area of the PE filtration membrane is fixed; therefore, the time required to achieve equilibrium for the applied flow rate is nearly constant. 

However, the filtration performance was different for the PE membranes coated with EO1V and EO2V with different deposition times. As shown in Table 2, all the flux nearly reached zero at 40 mins, and the thickness of the filtrate cakes differed. The analyses on the cross-section of PE membranes after filtrating CHO cells visualized by SEM provided information about the thickness of the accumulated filtrate cake and the effects of fouling resistance due to the surface modifications. The thickness of filtrate cake on pristine PE was 183.3 μm (Table 2, and Appendix A). The deposition of EO1V and EO2V showed effective resistance to the adhesion of CHO cells and cell solution content; therefore, the thickness of cake reduced to the minimum value of 69.3 μm and 67.9 μm on 30W-pp-EO1V/PE and 30W-pp-EO2V/PE, correspondingly. 

Zeta potential was applied to measure the surface charges of the plasma polymerized filtration membrane because it was reported to be one of the very important parameters that provides the fouling resistance. The pristine PE filtration membrane carried a negative surface charge of −59.0 ± 1.0 mV.

By depositing EO1V and EO2V on PE, the surface charge turned dramatically from negative toward positive, such that the surface potential was −4.0 ± 1.33 mV, and −4.9 ± 0.4 mV on 10W-ppEO1V/PE and pp50W-EO1V/PE, respectively (Figure 5a). It is noted that a value of ~4.0 mV of surface potential was found on all ppEO1V/PE and ppEO2V/PE, regardless of the applied plasma power, indicating that the surface charges can be dictated by the top layer of the plasma deposited thin film (Figure 5a,b). 

The fouling resistance experiments were conducted by the protein adsorption using bovine serum albumin (BSA) on pristine PE and ppEO1V/PE and ppEO2V/PE. On the pristine PE filtration membrane, more than 1186 μg/cm^2^ of BSA was found. A significant reduction in the amount of the adsorbed BSA, less than 820 μg/cm^2^, was found on pp-EO1V/PE and pp-EO2V/PE (Figure 5c,d). 

To show the efficacy of the surface modification by plasma polymerization, the reusability of the PE filtration membranes was evaluated by measuring the flux recovery ratio using NaOH solution and water. The flux recovery of pristine PE was 76.6% and 90.5% using water and NaOH solution (Figure 6a,b). For the EO1V and EO2V modified PE membranes, the applied power of plasma polymerization was set at 30 W according to the optimized results of WCA, zeta potential, and BSA adsorption.

It was found that more than 17% of flux recovery was found on 30-pp-EO1V/PE and 30-pp-EO2V/PE when compared with that on the pristine PE (Figure 6a) by flushing with water. On other hand, only 4% of flux recovery improvement was found on the plasma polymer deposited PE by using NaOH solution (Figure 6b). The results indicate that plasma polymerization of EO1V and EO2V effectively promoted the resistance to the adhesion of CHO cells not only for the first filtration but also for the possibility of reusing the filtration membrane by simply washing with water. 

## 4. Discussion

The target application of this study is to prolong the lifetime of the PE filtration membrane for the separation of CHO cells with smaller compounds secreted by CHO cells. The pp-EO1V/PE and pp-EO2V/PE modified filtration membranes were applied to evaluate the filtration performance against Chinese hamster Ovary cells. The improvement of filtration results on EO1V/PE and EO2V/PE was most probably due to the improvement of surface hydrophilicity for the applied power under 30 W. The lower filtration results at a higher applied power could be accounted for by the thicker deposition thin films of EO1V and EO2V on PE, which may cause the reduction in the size of the pores of the filtration membranes. It was found that the thickness of the filtrate cakes revealed the same trend; therefore, the filtration performance at 30 W is the optimal applied power for plasma polymerization of EO1V and EO2V in order to obtain the best filtration performance.

The relative positive surface charges of ppEO1V/PE and ppEO2V/PE were due to the incorporation of the ethylene glycol (-C-C-O) moieties from EO1V and EO2V, which were reported to carry neutral surface charges that covered the PE filtration membranes uniformly. It is noted that the results of static adsorption of BSA were similar to that of surface zeta potential and WCA, which show significant differences in surface properties among the pristine PE and the plasma polymer deposited PE. Moreover, the integration of all the acquired results implies that the weak applied power of plasma polymerization was sufficient to create thin films with good fouling properties, which is an important characteristic of the plasma polymerization technique.

The importance of this study can be addressed by comparing it with previous works. Rahimpour et al. has grafted acrylic acid and 2-hydroxyethylmethacrylate (HEMA), 2,4-phenylenediamine (PDA), and ethylene diamine (EDA) on polyvinylidene fluoride (PVDF) and found that the flux increased from the original 27 kg/(m^2^.hr) to 29, 31, 29, and 31 kg/(m^2^.hr), respectively. The increase of the flux was 7–14%—obtained by the grafting modifications [19]. In addition, Zou et al. has applied plasma polymerization to deposit a polyethylene glycol (PEG)-like hydrophilic polymer, triglyme, to reduce the fouling of reverse osmosis (RO) membrane. The filtration results show that there was a 5% decline in membrane permeability for the plasma-treated membrane when compared with the untreated one [14]. The highest permeability was about 1.0 L/m^2^.hr.bar. Previous works have shown that limited improvement, or even worse flux, can be obtained by the modifications. In this study, the flux increased from 175–225 L/ m^2^.hr, which accounted for more than 28% of flux increment. There is also literature that has reported to prepare EO1V and EO2V thin films individually [20,21] or simultaneously [22]. However, the EO1V and EO2V deposition was not applied to modify the filtration membranes. For example, Choi et al. applied capacitively coupled plasma chemical vapor to prepare patterned EO2V thin films to acquire FITC-tagged IgG (Immunoglobulin G) protein arrays [20]. The research group of Al-Hamarneh prepared plasma polymerized EO2V thin films and provided a full spectrum of analyses [21]. Wu et al. investigated the impacts of deposition power on the surface density of EO units by depositing EO1V and EO2V thin films, which resisted the adsorption of albumin and fibrinogen [22]. It could be summarized that most of the previous works did not apply the prepared plasma polymerized EO1V and EO2V thin films for industry applications such as cell filtration and separation. This allows us to address the significance of this specific work, which prepared antifouling filtration membranes to prolong the lifetime of membranes to separate mammalian cells with the filtrates.

## 5. Conclusions

Surface modification on PE filtration membranes was performed by plasma polymerization using EO1V and EO2V as precursors to incorporate ethylene glycol moieties to provide resistance to the adsorption of biomolecules. In this particular study, the resistance to mammalian cells, Chinese hamster ovary (CHO) cells, was evaluated because the pharmaceutical industries required the production of active compounds using CHO cells. The results show that the deposition of plasma polymers using EO1V and EO2V allowed the effective improvement of the filtration performance without using any toxic chemicals and could be facilitated within a very short modification time. The plasma parameters were optimized by the physico-chemical properties of the deposited ppEO1V and ppEO2V; therefore, the results of WCA confirmed the obtained results of zeta potential and protein adsorption. Moreover, the deposition of EO-containing plasma polymers resulted in larger flux and higher accumulation of weight on the PE filtration membranes and therefore prolonged the lifetime of the filtration membrane, which can therefore reduce the costs of membranes and the overall process. 

The mechanism for the production of the resistance to fouling is mainly due to the incorporation of the ethylene glycol moieties of EO1V and EO2V precursors as a result of plasma polymerization. The functionalities of the EO1V and EO2V precursors were verified by both FTIR and ESCA, which caused the increases in hydrophilicity and surface charges, which revealed the repulsion effects in both CHO cells and BSA. The outcomes of the increased filtration flux and accumulated filtrate weight on pp-EO1V/PE and pp-EO2V/PE confirmed the successes of the surface modifications. 

The overall results show that the plasma polymerization using precursors containing ethylene glycol (EG) moieties provides fouling resistance to both protein and CHO cell adhesion and great potential for the perfusion experiments in protein production at an industrial scale.

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
