# Peer review of "Surface Antifouling Modification on Polyethylene Filtration Membranes by Plasma Polymerization"

_materials, 2020, doi:10.3390/ma13215020_

Round 1

Reviewer 1 Report

Dear author,

    The manuscript materials-964286, entitled ‘Surface Antifouing Modification on Polyethylene Filtration Membranes by Plasma Polymerization’ presents a new method, namely plasma polymerization to deposit antifouling thin films onto polyethylene membranes using two types of EO-containing precursors, monoethylene vinyl ether (EO1V) and diethylene glycol vinyl ether (EO2V), under different applied power, in order to evaluate the outcomes of biofouling toward proteins and mammalian cells. Furthermore, the physico-chemical properties, including surface wettability, chemical functional groups, and surface charges of the deposited thin films, were evaluated in order to clarify the mechanism for fouling resistance.

    Characterization methods, such as: FTIR (Fourier transform Infrared), SEM (scanning electron microscopy), and X-ray photoelectron spectroscopy (XPS) analyses, water contact angle (WCA) measurements and permeation experiments were used for verifying the successful deposition of the plasma polymers and to evaluate the change of hydrophilicity and fouling resistance of these plasma polymerized films.

    The results are presented well, supported by adequate images and graphs (with good resolution) revealing the infos needed to sustain the conclusions drown by the authors.

Overall, the manuscript is well written, with many techniques / methods described for the characterization of the polymerized films, with good correlation between the obtained results.

However, there are some clarifications that MUST be made in the experimental set-up / working conditions part of the manuscript:

1) on page 2, third paragraph: ‘2.2 Plasma polymerization’ some infos upon the plasma polymerization conditions MUST be made:

a) a schematic with the plasma reactor should be inserted in order to visualize the plasma region for polymerization

b) the total / partial working pressure MUST be mentioned, as the sentence on the lines 79-81: ‘The deposition of precursors was set at 100 mTorr, 20 sccm, with heating temperature at 100 °C, for 60 min.’is rather confusing. What was the total pressure during the plasma polymerization process (is it 100 mTorr?); the flow rate of the both precursors was identical ( ~20 sccm?).

c) the working gas was xxx? Is it in flow in the reactor chamber? If so, the flow rate is? What is the discharge type? Is it a glow discharge? Please put these infos into the manuscript.

d) How was the temperature measured during the deposition process and were it was measured? Is it near the electrodes, of in the chamber? Please specify this aspect.

    The above information’s must be added, since starting with the title of the manuscript plasma polymerization is the ‘novelty’ for obtaining these films with remarkable properties.

    The authors find that the plasma polymerized films were hydrophilic (WCA below 20˚) and with low roughness (Rms (nm): 0.2 to 0.5), with an increased content of oxygen (from FTIR: 16 % to 28% more O than the PE substrate, and by ESCA from 16 % to 28% more O).  The plasma power influence upon the oxidation of the deposited films was also observed. The results presented by the authors showed that the deposition of plasma polymers using EO1V and EO2V allowed to effectively improve the filtration performance without using any toxic chemicals and could be facilitated within very short modification time.

    The conclusions are in brief but sustained (most of the conclusions) by the experimental findings.

    In the end, based on their experimental findings, the authors concluded that the plasma polymerization using precursors containing ethylene glycol (EG) moieties provide fouling resistance to both protein and CHO cell adhesion and might have great potential for the perfusion experiments in the procedure of protein production in industrial scale.

    I propose that this manuscript can be accounted for publishing after MINOR REVISIONS in the journal MATERIALS.

Author Response

The reviewer commented that the manuscript presents a new method, namely plasma polymerization to deposit antifouling thin films onto polyethylene membranes using two types of EO-containing precursors, monoethylene vinyl ether (EO1V) and diethylene glycol vinyl ether (EO2V), under different applied power, in order to evaluate the outcomes of biofouling toward proteins and mammalian cells. The plasma power influence upon the oxidation of the deposited films was also observed. The results presented by the authors showed that the deposition of plasma polymers using EO1V and EO2V allowed to effectively improve the filtration performance without using any toxic chemicals and could be facilitated within very short modification time.The reviewer commented that the conclusions are in brief but sustained (most of the conclusions) by the experimental findings. The reviewer mentioned that the results are well presented and the manuscript is well written. However, there are some clarifications that “must” be made in the experimental set-up/working conditions part of the manuscript.

We thank the reviewer for the positive comments. The responses to the questions of the reviewer are replied in a point-to-point to manner as following:

Comment 1. The reviewer commented that some info upon the plasma polymerization conditions must be made on page 2:

  • a schematic with the plasma reactor should be inserted in order to visualize the plasma region for polymerization

We appreciated the suggestion from the reviewer and have added the schematic diagram of the plasma reactor used for plasma polymerization in Scheme 1 in the revised manuscript.

(Line 75) … The schematic diagram of the plasma reactor is shown in Scheme 1….

  • the total / partial working pressure, the type of gas and gas flow rate, the temperature during the deposition process as well as the glow discharge must be mentioned,

We appreciated the suggestion from the reviewer and have added the information of total and working pressure in the revised manuscript.

(Lines 79 - 81) … . The working pressure for the deposition of precursors was set at 100 mTorr, with flow rate of precursors at 20 sccm, with heating temperature at 100 °C, for 60 min. The generated plasmas is glow discharge.. …

Reviewer 2 Report

In the submitted manuscript, the authors present data of surface modification on PE membranes by plasma polymerization to provide the anti-fouling ability. The topic of this in vitro study falls within the scope of the journal and it is of scientific interest. The manuscript is well written.

Author Response

We thank the reviewer for the positive comments and will continue to work into the depths in this field.

Reviewer 3 Report

The comments are included in the attached file.

Author Response

The reviewer commented that the manuscript analyzes the modification of polyethylene membranes by plasma polymerization of EO1V and EO2V. The aim of the treatment is the improvement of the performance of the filtration membranes with foulant feeds. The topic is interesting but the manuscript must be improved to be accepted for publication.  The referee also pointed out that the authors should explain the novelty of their work when compared to previously published paper in this topic, which have not been included as references:

  1. Yuliang J. Wu, Richard B. Timmons, James S. Jen, Frank E. Molock (2000). Non-fouling surfaces produced by gas phase pulsed plasma polymerization of an ultra low molecular weight ethylene oxide containing monomer. Colloids and Surfaces B: Biointerfaces 18 (2000) 235–248
  2. Pedrow, I. Alhamarneh, S, Goheen (2007). Plasma-assisted Grafting of Polyethylene Glycol (PEG) to Solid Substrates. IEEE 34th International Conference on Plasma Science (ICOPS), Albuquerque, NM, 2007, pp. 564-564, doi: 10.1109/PPPS.2007.4345870.
  3. Changrok Choi, Donggeun Jung, DaeWon Moon, Tae Geol Lee (2011). Surface analysis of protein-resistant, plasma polymerized ethylene glycol thin films. Surf. Interface Anal. 2011, 43, 331–335.
  4. Ibrahim Al-Hamarneh, Patrick Pedrow, Asma Eskhan, Nehal Abu-Lail (2013). Synthesis and characterization of di(ethylene glycol) vinyl ether films deposited by atmospheric pressure corona discharge plasma. Surface & Coatings Technology 234 (2013) 33–41.

The reviewer also suggested to modify some points which we have prepared the response in a point-to-point manner as following.:

Comment 1. The reviewer mentioned that the authors should further compare the performance of the membranes. For example, the permeability values of the different membranes for water filtration should be introduced.

We appreciated the comments from the reviewer. By surveying previous works, Rahimpour et al grafted acrylic acid and 2-hydroxyethylmethacrylate (HEMA), 2,4-phenylenediamine (PDA), and ethylene diamine (EDA) on PVDF (poly(vinylidene fluoride) and found that the flux increased from the original 27 kg/(m2.hr) to 29, 31, 29, and 31 kg/(m2.hr), respectively, The increase of the flux is 7 – 14 % by the grafting modifications [1]. In addition, Zou et al have applied plasma polymerization to deposited a polyethylene glycol (PEG)-like hydrophilic polymer, triglyme, to reduce the fouling of RO membrane. The filtration results showed that there is a 5 % decline in membrane permeativity for the plasma-treated membrane when compared with the untreated one [2]. The highest permeability if about 1.0 L/m2žhr.bar. The previous works showed that limited improvement of even worse flux can be obtained by the modifications. In this manuscript, the flux increased from 175 – 225 L/ m2.hr which accounted more than 28 % of flux increment. There was also literature reported to prepare EO1V and EO2V thin films individually [3-4] or simultaneously [5]. However, the EO1V and EO2V deposition was not applied for modifying the filtration membranes. For example, Choi et. al. has applied capacitively coupled plasma chemical vapor to prepare patterned EO2V thin films to acquire FITC-tagged IgG protein arrays [3]. The research group of Al-Hamarneh has prepared plasma polymerized EO2V thin films and provided a full spectrum of analyses [4]. Wu et al have investigated the impacts of deposition power on the surface density of EO units by depositing EO1V and EO2V thin films which resisted the adsorption of albumin and fibrinogen [5]. As shown that most of the previous works did not really apply the prepared plasma polymerized EO1V and EO2V thin films for real works such as cell filtration and separation works. This allows to address the importance of this specific work which prepared antifouling filtration membranes to prolong the lifetime of membranes to separate mammalian cells with the filtrates.

[1] Rahimpour, S. S .Madaeni, S. Zereshki, Y. Mansourpanah, Preparation and characterization of modified nano-porous PVDF membrane with high antifouling property using UV photo-grafting, Applied Surface Science, 225 (16), 2009, 7455-7461
[2] L. Zou, I. Vidalis, D. Steele, A. Michelmore, S. P. Low, J. Q. J. C. Verberk, Surface hydrophilic modification of RO membranes by plasma polymerization for low organic fouling, Journal of Membrane Science, 369 (1-2), 2011, 420-428
[3] Changrok Choi, Donggeun Jung, DaeWon Moon, Tae Geol Lee, Surface analysis of protein-resistant, plasma polymerized ethylene glycol thin films. Surf. Interface Anal. 43, 2011, 331–335.
[4] Ibrahim Al-Hamarneh, Patrick Pedrow, Asma Eskhan, Nehal Abu-Lail, Synthesis and characterization of di(ethylene glycol) vinyl ether films deposited by atmospheric pressure corona discharge plasma. Surface & Coatings Technology 234, 2013, 33–41.
[5] Yuliang J. Wu, Richard B. Timmons, James S. Jen, Frank E. Molock (2000). Non-fouling surfaces produced by gas phase pulsed plasma polymerization of an ultra low molecular weight ethylene oxide containing monomer. Colloids and Surfaces B: Biointerfaces 18, 2000, 235–248

Comment 2. The reviewer suggested that, from the graphs in Figure 4, once 40 minutes of test have been completed, the performance of all the membranes is quite similar, it is needed to be explained by the authors.

We appreciated the comments from the reviewer and have added the explanation in the revised manuscript.

(Lines 268 - 271) It is noted that all the filtration results reached the same flux quantity after 40 min of filtration, indicating that the termination of the filtration and equilibrium filtered cakes were obtained, which related closely to the amount of the cell solutions applied in this study.

Comment 3. The reviewer commented that the language of the paper is very poor and must be clearly improved. In addition, the authors use expressions without previous introduction, such as O.D. in Line 131 or WCA in line 140.

We thank the reviewer for the comments and have carefully proof read the manuscript with all the corrections highlighted in the revised manuscript. The full names for the abbreviation OD and WCA were also provided in the revised manuscript.

(Lines 145 - 146) Cell solution was diluted until the O.D. (optical density) value equaled to 0.3 prior to the filtration tests…

(Lines 154 - 155) … characterized by the deposited thickness, WCAs (water contact angles), zeta potential,…

Comment 4. The reviewer found that Figure 1b is refereed by the authors, but it does not exist. The captions of Figures 2 and 3 are confusing.

We thank the reviewer for the corrections and have modified both the figures and captions in Figure 1. We have modified the figure captions for Figures 2-3 to avoid confusing in the revised manuscript.

Figure 1. The thickness of the plasma polymerizated thin films under different applied power using (i) EO1V and (ii) EO2V precursors. (Plasma deposition pressure: 100 mtorr, flow rate of precursors: 15 sccm, deposition time: 0.5 h)

Figure 2. The surface morphology visualized by SEM: (a) on pristine PE, on EO1V/PE at different applied plasma power, (b) 10- EO1V/PE, (c) 20-EO1V/PE, (d) 30-EO1V/PE, (e) 40-EO1V/PE, (f) 50-EO1V/PE; and on EO2V/PE at different applied plasma power: (g) 10-EO2V/PE, (h) 20-EO2V/PE, (i) 30-EO2V/PE, (j) 40-EO2V/PE, (k) 50-EO2V/PE. (Plasma deposition pressure: 100 mtorr, flow rate of precursors: 15 sccm, deposition time: 0.5 h)

Figure 3. Surface chemical functionalities analyzed by FTIR on (a) pp-EO1V/PE, and (b) pp-EO2V/PE as a function of the plasma applied power. (Plasma deposition pressure: 100 mtorr, flow rate of precursors: 15 sccm, deposition time: 0.5 h)

Reviewer 4 Report

The manuscript describe the surface modification of polyethylene filtration membranes by plasma polymerization. The authors used two hydrophilic precursors: ethylene oxide vinyl ether and diethylene oxide vinyl ether to improve the antifouling properties of the polyethylene membranes against mammalian cells (Chinese Hamster Ovary) and proteins (bovine serum albumin).

The manuscript is fit within the scope of the Materials and is of interest to its readers. Therefore, the manuscript may be accepted for the publication by the journal, after the minor revision.

The following issues should be addressed:

  • Title - Please correct the title...
  • Line 75 - RF - please provide the full name of the abbreviation
  • Line 138 - Please change to: "The effects of..."
  • Line 140 - WCA - Please mention the full name before using the abbreviation
  • Figure 1 - The authors are suggested to use the same nomenclature for the figure: either (i) and (ii) or (a) and (b)...
  • Line 309 - It is suggested to check the sentence:  "It was is found that..."
  • The authors are suggested to reduce the conclusions and make this section more concise.
  • Also, it is suggested to proof-read the manuscript to avoid some minor mistakes...

Author Response

The reviewer mentioned that the manuscript describes the surface modification of polyethylene filtration membranes by plasma polymerization. The authors used two hydrophilic precursors: ethylene oxide vinyl ether and diethylene oxide vinyl ether to improve the antifouling properties of the polyethylene membranes against mammalian cells (Chinese Hamster Ovary) and proteins (bovine serum albumin). The referee also commented that the manuscript is fit within the scope of the Materials and is of interest to its readers. Therefore, the manuscript may be accepted for the publication by the journal, after the minor revision.

We acknowledged the suggestions and comments from the reviewer and have prepared the responses in a point-to-point manner as following:

Comment 1. The reviewer suggested to correct the title

We thank the reviewer for the corrections and we have modified the title of the manuscript in the revised manuscript as “Surface Antifouling Modification on Polyethylene Filtration Membranes by Plasma Polymerization”.

Comment 2. The reviewer suggested to provide the full name of the abbreviation of RF at Line 75.

We thank the reviewer for the suggestions and we have added the full names for the abbreviation at line 75 in the revised manuscript.

(lines 73 - 75) In brief, the plasma system is composed of three main parts: (1) a reaction chamber, (2) a RF (radio-frequency) generator

Comment 3. The reviewer suggested to change to: "The effects of..." at Line 138.

We thank the reviewer for the corrections and we have modified the paragraph at line 138 in the revised manuscript.

(lines 152 - 153) The effects of plasma polymerization deposition on polyethylene membranes using two ethylene oxide containing precursors,

Comment 4. The reviewer suggested to mention the full name before using the abbreviation at Line 140.

We thank the reviewer for the suggestions and we have added the full names for the abbreviation at line 140 in the revised manuscript.

(lines 153 - 157) … EO1V (ethylene oxide vinyl ether) and EO2V (diethylene oxide vinyl ether), were characterized by the deposited thickness, WCAs (water contact angles), zeta potential, and SEM. The filtration performance was evaluated by directly filtrating CHO cells. The surface fouling properties were quantified by the amount of adsorbed protein using bovine serum albumin (BSA).

Comment 5. For Figure 1, the reviewer suggested to use the same nomenclature for the figure: either (i) and (ii) or (a) and (b)...mention the full name before using the abbreviation

We thank the reviewer for the correction and we have modified the incorrect nomenclature for the figures in the revised manuscript.

Figure 1. The thickness of the plasma polymerizated thin films under different applied power using (i) EO1V and (ii) EO2V precursors. (Plasma deposition pressure: 100 mtorr, flow rate of precursors: 15 sccm, deposition time: 0.5 h)

Comment 6. The reviewer suggested to check the sentence:  "It was is found that..." at Line 309.

We thank the reviewer for the correction and we have modified the incorrect sentence in the revised manuscript.

(lines 327 - 328) … It was found that the thickness of the filtrate cakes revealed the same trend such that the filtration performance at 30 W

Comment 7. The reviewer suggested to o reduce the conclusions and make this section more concise.

We thank the reviewer for the suggestion and we have proof read the manuscript carefully with all the corrections highlighted in red color.

Round 2

Reviewer 3 Report

The comments are included in the attached Word file.

Author Response

Manuscript: Surface Antifouling Modification on Polyethylene Filtration Membranes by Plasma Polymerization (materials-964286 R2)

Responses to Reviewers' comments

3rd Reviewer

Comment 1. The reviewer mentioned that the authors have prepared an improved version of the manuscript. However, the main aspects during the first review were not answered satisfactorily. On the one hand, the reviewer also suggested to incorporate the further justification of the novelty aspects of this paper when compared to previous work in the responses to the reviewer into the revised manuscript.

We appreciated the reviewer for the important suggestions and have implemented the revised manuscript by adding the information about the novelty aspects of this paper when compared with previous works.

(Lines 339 - 361)

The importance of this study can be addressed by comparing with previous works. Rahimpour et. al. has grafted acrylic acid and 2-hydroxyethylmethacrylate (HEMA), 2,4-phenylenediamine (PDA), and ethylene diamine (EDA) on PVDF (polyvinylidene fluoride) and found that the flux increased from the original 27 kg/(m2.hr) to 29, 31, 29, and 31 kg/(m2.hr), respectively. The increase of the flux was 7 – 14 % by the grafting modifications [19]. In addition, Zou et. al. has applied plasma polymerization to deposit a polyethylene glycol (PEG)-like hydrophilic polymer, triglyme, to reduce the fouling of RO membrane. The filtration results showed that there was a 5 % decline in membrane permeability for the plasma-treated membrane when compared with the untreated one [20]. The highest permeability was about 1.0 L/m2žhr.bar. The previous works showed that limited improvement or even worse flux can be obtained by the modifications. In this study, the flux increased from 175 – 225 L/ m2.hr which accounted for more than 28 % of flux increment. There was also literature that reported to prepare EO1V and EO2V thin films individually [21-22] or simultaneously [23]. However, the EO1V and EO2V deposition was not applied for modifying the filtration membranes. For example, Choi et. al. has applied capacitively coupled plasma chemical vapor to prepare patterned EO2V thin films to acquire FITC-tagged IgG protein arrays [21]. The research group of Al-Hamarneh has prepared plasma polymerized EO2V thin films and provided a full spectrum of analyses [22]. Wu et. al. has investigated the impacts of deposition power on the surface density of EO units by depositing EO1V and EO2V thin films which resisted the adsorption of albumin and fibrinogen [23]. It could be summarized that most of the previous works did not apply the prepared plasma polymerized EO1V and EO2V thin films for industry such as cell filtration and separation. This allows to address the significance of this specific work which prepared antifouling filtration membranes to prolong the lifetime of membranes to separate mammalian cells with the filtrates.

Comment 2. The reviewer also commented that the filtration performance of the membranes still needs to be further explained. The authors, when asked about the results in Figure 4, consider that “all the filtration results reached the same flux quantity after 40 min of filtration, indicating that the termination of the filtration and equilibrium filtered cakes were obtained, which related closely to the amount of the cell solutions applied in this study”. Do the authors consider the time required to achieve this equilibrium is constant for all the membranes and does not depend on the membrane permeability?

We appreciated the reviewer for the important question about the filtration performance. Due to the limit of the crossed-flow filtration system (as shown in Supplementary Information Figure S4), the surface area of the PE filtration membrane is fixed, therefore the time required to achieve equilibrium for the applied flow rate is nearly constant. However, the filtration performance was different on the PE membranes coated with EO1V and EO2V with different deposition time. As Table 2 showed, even all the flux reached nearly zero at 40 mins, the thickness of the filtrate cakes differed. We have added the explanation for the filtration performance in the revised manuscript.

(Lines 266 - 276)

The filtration results clearly showed that the slower reduction of and later blockage of EO1V/PE and pp-EO2V/PE when compared with the pristine PE (Figures 4c and 4d). It is noted that all the filtration results reached the near zero flux quantity after 40 min of filtration, indicating that the termination of the filtration and equilibrium filtered cakes were obtained, which related closely to the amount of the cell solutions applied in this study. Due to the limit of the crossed-flow filtration system (as shown in Figure S3 in supporting information), the surface area of the PE filtration membrane is fixed, therefore the time required to achieve equilibrium for the applied flow rate is nearly constant.

However, the filtration performance was different on the PE membranes coated with EO1V and EO2V with different deposition time. As shown in Table 2, even all the flux reached nearly zero at 40 mins, the thickness of the filtrate cakes differed.

Comment 3. In the previous review the reviewer, the reviewer has questions for further information about the performance of all the membranes when filtering just water (permeate fluxes) which was not answered from the previous revision

We thank the reviewer for the important question and we were sorry that we didn’t answer this question previously. For this particular industrial cooperation, we have been focused on filtrating the CHO (Chinese hamster ovary) cell solution to prolong the filtration membrane, therefore the only target is the cell solutions and to evaluate how long the PE membrane can be used. Therefore, we were not able to provide the results for the permeate fluxes because we didn’t perform the experiments by filtrating just water. However, prior to the filtration experiment, the PE membranes were immersed in DI water for at least 30 mins. For the future work, we will conduct the water filtration to obtain the permeate fluxes. We hope this is acceptable.

Comment 4. In addition, the reviewer suggested that the language must still be improved (Line 81 plasmas is, Line 97 poymierzation...) and some expressions must be explained (Line 81 sccm without reference to standard cubic centimeter per minute).

We thank the reviewer for the correction and we have modified the incorrect sentence in the revised manuscript.

(lines 80 - 81) …, with flow rate of precursors at 20 sccm (standard cubic centimeters per minute),…

(line 82) …, The generated plasma is a glow discharge…

(caption for scheme 1) Scheme 1. The plasma reactor for plasma polymerization, composed by: (1) reactor, (2) mechanical pump, (3) mass flow rate controllers, (4) RF generator, and (5) matching box.

Comment 5. The reviewer consider the manuscript must be improved to be accepted for publication

We thank the reviewer for the comments and have carefully proof read the manuscript with all the corrections highlighted in red in the revised manuscript.
